# Effect of table tennis balls with different materials and structures on the hardness and elasticity

**Yunfei Lu**[1], **Jie Ren**[1], **Jun Wang**[2], **Yan Wang**[1]*

1 China Table Tennis College, Shanghai University of Sport, Shanghai, China, 2 Univ. Grenoble Alpes, GIPSA-Lab, CNRS, Grenoble, France

* 1362284407@qq.com

**Data Availability Statement:** All relevant data are within the manuscript and its Supporting Information files.

**Funding:** The author(s) received no specific funding for this work.

## Abstract

The new material introduced by the ITTF in 2014 for table tennis balls has attracted significant attention from players and coaches. Changes in both material selection and production procedures are likely to have affected the static performance of the ball. However, the raw data regarding the elasticity and hardness of these new material balls, encompassing various brands and structures, often lacks practical information crucial for players' rapid adaptation and daily training. The static properties tested in this study were provided by the ITTF, covering both hardness and elasticity. Based on computed variables, this study revealed that the hardness of seam balls at the equator was not consistently higher than that at the pole. Additionally, the study confirmed that the hardness and bounce height of new material balls exceeded those of celluloid. Furthermore, correlation analysis was conducted to examine the relationship between these two properties, revealing a significant correlation between the hardness of seamless balls and their elasticity. This study provides an analysis of the static performance of various types of new material balls, aiding players and coaches in better understanding official event balls and offering a theoretical foundation for the formulation of diverse training and game strategies.

## Introduction

Competitive table tennis underwent several equipment changes over the last two decades that have affected gameplay. Recently in 2014, new plastic balls were introduced in all events sanctioned by the International Table Tennis Federation (ITTF). The old celluloid balls used for over 140 years were phased out due to environmental, safety, and cost issues [1]. Demand for substitute non-toxic, non-hazardous, and environmental balls for celluloid balls has heightened the developments in materials and production processes. The ITTF equipment committee conducted several mechanical tests (Technical leaflet T3) [2] for manufacturers to investigate the vertical bounce height and the compressive strain to determine the properties, such as elasticity and hardness.

These celluloid-free balls were identically-named 'new material balls' and divided into 'seam' and 'seamless' depending on producing methods. The seam balls were mainly

**Competing interests:** The authors have declared that no competing interests exist.

compression-molded by two hemisphere ball shells, in which plastic sheets were injected or stamped as shells, and seams would be polished smooth. Unlike the seam balls, seamless balls are entirely hollow and thin-walled ball shells, which were injection molded by a ball mold.

Since being introduced in 2014, ITTF has approved 79 new material balls, including 64 seam and 15 seamless balls [3]. As shown in Table 1, the tournament organizers would choose different brands and structures of balls depending on the sponsors and other circumstances.

Even though the specifications for the new materials balls were similar to that of old celluloid balls, it was expected that there would be some differences between them. Given table tennis balls' lightweight and low-density characteristics, any changes in diameter and roundness to a ball are likely to affect its flight trajectory and the interactions between the ball, table, and racket. A study comparing a new ball and the old celluloid ball is that of Tang et al. [4], in which the initial speed of the new ball was 1–2% less, and the ball spin was 5–20% less than for the old ball. The study of Marcus et al. [5] also reached similar conclusions, further finding a greater speed increment (0.69%) and smaller spin decrement (0.19%) than celluloid ball after post ball-table impact. The decrease in ball speed could enhance the spectator experience of table tennis, particularly considering the decline in media interest in the sport, especially outside of Asia. This decline is partly attributed to the fast-paced nature of the game, making it challenging for viewers to track the ball [6]. The transition from the old 38-mm ball to the larger 40-mm ball has resulted in reduced velocity and spin, as noted by several researchers [7–9]. This reduction stems from the increased air resistance due to the larger cross-sectional area of the new ball. Additionally, the redistribution of mass away from the center of the larger ball increases its inertial moment, further diminishing spin. Furthermore, Xie et al. found that the 40-mm ball exhibits decreased speed and rotation compared to the 38-mm ball, with the reduction in speed outweighing the loss of rotation. This effect is attributed to the increased air resistance encountered by the larger ball during flight. However, the degree of speed reduction varies among players and skill levels. Their study revealed that the reduction in speed for forehand smashes ranged from 0.0% to 7.9%, while the reduction in forehand high-loop spin ranged from 2.0% to 7.7% [10].

In a subjective survey of new plastic balls, Zhang et al. [11] reported that a great majority of players thought that new balls have a more stable bounce and different hitting sound compared to celluloid. An observational study conducted by Wei et al. [12] found that players in the Chinese national team scored significantly lower in the attack after receiving section when the new plastics ball was used. Furthermore, despite less impact in the services and stalemate stages, specific techniques and tactics were also affected. Since players have accumulated information regarding the behavior of celluloid balls, considering that this behavior changes due to the new material, the player's prediction scales need to be adjusted accordingly [13]. The main challenge faced by many trainers and coaches is how to adapt to official balls quickly, to perform best competitive state.

**Table 1. Approved balls in events.**

| Brand | Product | Type | Event |
|---|---|---|---|
| Butterfly | A40+ | With seam | 2018 World Table Tennis Championships |
| DF | V40+ | With seam | 2019 Budapest World Table Tennis Championships |
| DHS | D40+ | With seam | 2017–2020 Table tennis World Cup |
| DHS | DJ40+ | With seam | Tokyo 2020 Olympic Games |
| 729 | 40+ | Seamless | The 13th National Games of the People's Republic of China |
| Joola | Flash 40+ | Seamless | The designated European Games |

Note: DHS: Double Happiness, DF: Double Fish

A ball's elasticity and hardness are the most intuitive impression to the trainers. However, the performance of these properties is affected by the material, size, structures, and even brands. More recently, Yuki (2017) reported that the elastic recovery coefficient of the new material balls was higher than that of old celluloid balls. Experiments such as that conducted by Chen et al. (2014) have shown that greater hardness and elasticity were found in new material balls compared to celluloid. This finding was also reported by Xiao et al., in addition, they also found that seam balls have these higher properties than that seamless ball. Due to several basic properties, such as rotation, speed, striking point, and bounce height, have differences in performance with celluloid balls, Wang et al. (2021) [17] suggested trainers have to be more deeply familiar with the new material balls.

However, research on this subject up to now has been mostly restricted to the limited comparisons between the new material ball and celluloid balls, or individual seam and seamless ball. Few authors have been able to draw on much experimental research into the effect on ball's properties from different manufacturers. Players have a limited understanding of different kinds of balls, and few scientific experiments were conducted on them.

Therefore, the primary purpose of this study was to investigate the performance of different types (such as brands, structures, and materials) of table tennis balls. Mechanical tests using the T3 standard were performed to compare the elastic and hardness properties between new materials and celluloid, seam and seamless, different brands. Effective data and suggestions to evaluate the balls' characteristics will provide a theoretical foundation and reference for the development of differentiated training and game plans for various table tennis balls materials.

## Material and methods

### Samples

Nine types of table tennis balls approved by ITTF contained different brands and structures and were used in this study. Table 2 shows the basic information about the balls. Samples were randomly selected from each type of ball and were tested under the various conditions detailed below. According to the ITTF standard and the China national standards (GB/T 20045–2005) for 40mm table tennis balls, all samples were tested, and no significant differences were found in diameter, mass, and quality (Ranking).

### Method

The hardness and elasticity properties of the balls were tested according to ITTF, separately. Statistical significance between different structures (seam and seamless), test positions (polar and equatorial), and brands were analyzed using analysis of variance and t-test as appropriate.

**Table 2. Basic information of nine table tennis balls.**

| Structure | Brand | Model | Diameters (mm) |
|---|---|---|---|
| seam | DHS | Celluloid 40+ | 40 |
| | DHS | D40+ | 40 |
| | DHS | DJ40+ | 40 |
| | Nittaku | Premium 40+ | 40 |
| | DF | V40+ | 40 |
| | Butterfly | A40+ | 40 |
| Seamless | Yinhe | S40+ | 40 |
| | 729 | 40+ | 40 |
| | Joola | Flash 40+ | 40 |

Note: DHS: Double Happiness, DF: Double Fish

## Hardness test

Tests were performed on the dynamic uniaxial hardness testing machine (JSV-1000, ALGOL, Japan). As shown in Fig 1, the balls were fixed by a metal ball holder and extruded by a chuck in a vertical direction. The chuck was pulled at the displacement velocity of 10 mm/min by a metal rod with a diameter of 20 mm. The pressure and strain change can be accurately obtained by a high-precision mechanical sensor. The pressure values per 0.01mm of deformation were displayed on the monitor. The experiments were stopped when the value of pressure was higher than 50 N, or the deformation displacement exceeded 1mm. Nine samples were randomly selected from each type of ball, each sample was tested 3 times: the equator and two poles of seam balls were tested once, separately. The seamless balls were randomly placed and tested 3 times. The change between pressure and strain was regarded as a linear relationship, in according to the minor change in deformation during the test. Simple regression was analyzed the values of pressure-strain. Hardness coefficient K, defined as the regression coefficient, was used to judge the ball's hardness.

## Elasticity test

Fig 2 shows the elasticity tester (M0633, DHS, China) used in this study. The maximal height of the bounce was measured by dropping the balls from a height of 305 mm onto a windless metal plane. Select 15 random samples of each brand of ball, for each ball, the rebound height was calculated from the experimental image captured by a high-speed camera. *H*, which reflected the ball's elasticity, was taken as the average value of maximal height obtained over 3 repeated tests.

## Statistical analysis

Statistical analyses were conducted using SPSS (Version 22, IBM, USA). Descriptive statistics are average and standard error of mean (SEM). Data was prior analyzed by Kolmogorov-Smirnov test to confirm their normal distribution. Once validated, unpaired t-test were used to compare the differences in the hardness coefficients *K* of seams balls between pole and equators. One-way ANOVA was used ti test the variability in *K* and *H* between seam, seamless and celluloid balls. Linear correlation analysis was also performed between *K* and *H*. For all tests, a significance threshold set at P < 0.05 was chosen.

# Results

## Hardness

**Comparison of with seam balls poles and equator hardness.** Fig 3 shows the hardness coefficient at pole and equator of seam balls. As shown, what stand out is, hardness at equator of new material balls were significantly higher ($P < 0.001$) than that of celluloid ball. Furthermore, hardness at poles were also higher ($P < 0.001$) than the celluloid, except the Double Fish V40+ and DHS DJ40+, which showed new material balls exhibited quite different hardness than celluloid. Another interesting aspect of this graph is that significantly differences were found in DHS celluloid 40+ ($P = 0.006 < 0.01$), DHS D40+ ($P < 0.001$) and DHS DJ40+ ($P = 0.04 < 0.001$). This result indicated that the hardness at different positions of balls were also not same, might be caused in material and production process.

**Comparison of the hardness of seamless and with seam balls.** The average of hardness at pole and equator were calculated as the values of seam balls to compare with that of seamless balls. The mean values of hardness are presented in Fig 4. A one-way ANOVA revealed that

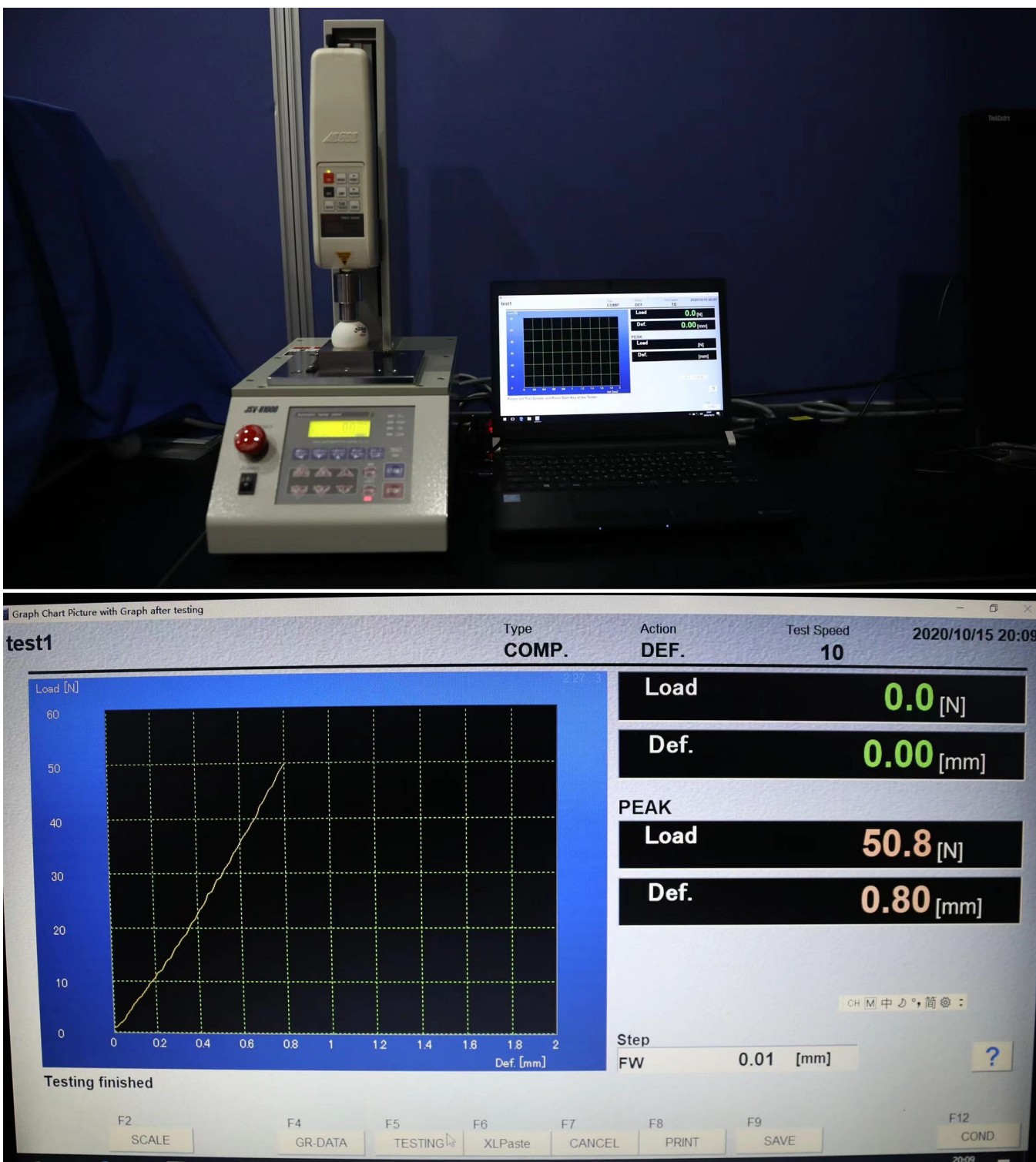

**Fig 1. JSV-H1000 table tennis hardness tester.**

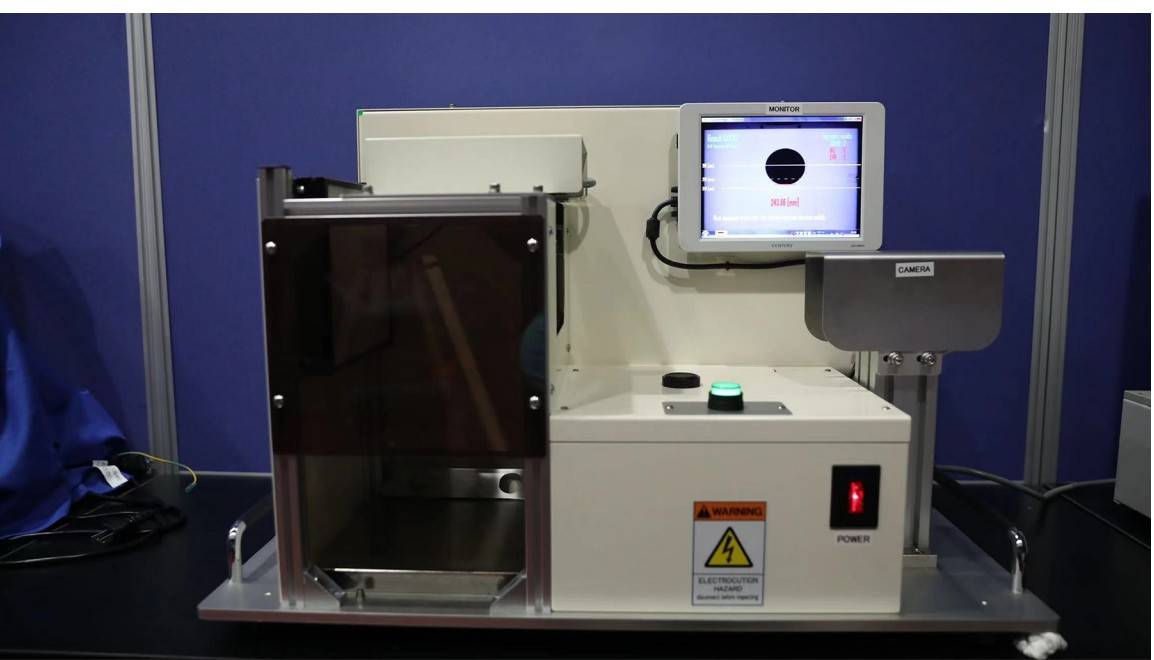

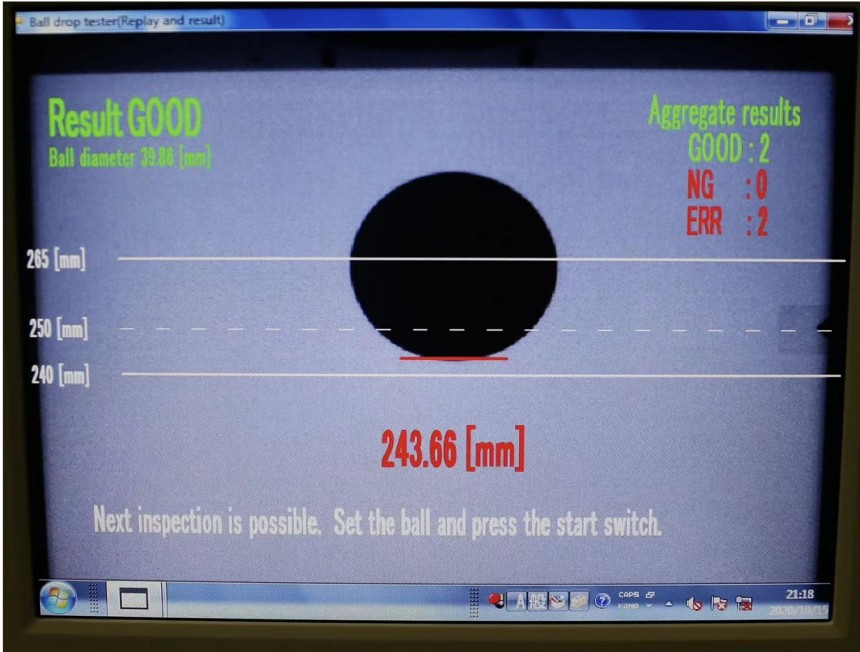

**Fig 2. M0633 table tennis ball elasticity tester.**

there was a significant main effect between nine types of tested balls ($F$ (8, 72) = 18.142, $P < 0.05$, $\eta_p^2 = 0.668$). As Table 3 shows, further independent sample t-test showed that hardness of 729 was higher than other two seamless balls. Comparison with seam balls, hardness of Butterfly A40+, Nittaku premium and DHS D40+ were significantly higher than that of total three seamless ball. Furthermore, what is striking about Fig 4 and Table 3 is all new material balls have higher hardness than celluloid ball.

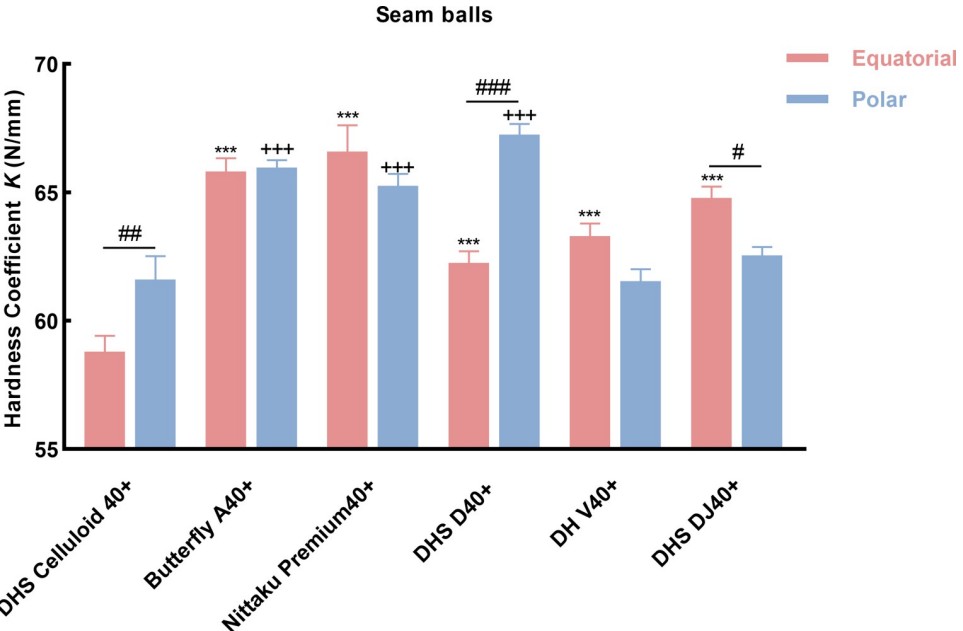

**Fig 3. Mean ± SEM of hardness coefficient at pole and equator of the seam balls.** (* Significant difference at equator compare with celluloid ball; +Significant difference at pole compare with celluloid ball; # Significant difference between pole and equator; *: $P < 0.05$; **: $P < 0.01$; ***: $P < 0.001$).

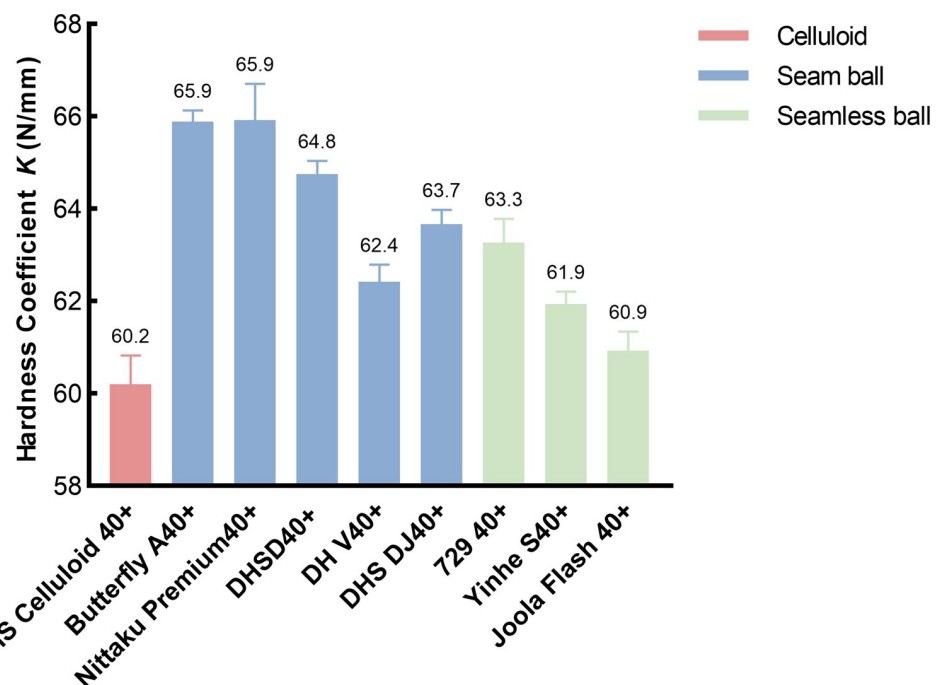

**Fig 4. Mean ± SEM of hardness coefficient of the seam and seamless balls.**

**Table 3. *P* values in Holm-Sidak's multiple comparison t-test in hardness.**

|  | Model | C 40+ | A40+ | P 40+ | D40+ | V40+ | DJ40+ | 40+ | S40+ |
|---|---|---|---|---|---|---|---|---|---|
| Seam | C 40+ | — |  |  |  |  |  |  |  |
|  | A40+ | <0.001*** | — |  |  |  |  |  |  |
|  | P 40+ | <0.001*** | 0.96 | — |  |  |  |  |  |
|  | D40+ | <0.001*** | 0.51 | 0.51 | — |  |  |  |  |
|  | V40+ | 0.02* | <0.001*** | <0.001*** | 0.01* | — |  |  |  |
|  | DJ40+ | <0.001*** | 0.02* | 0.02* | 0.52 | 0.46 | — |  |  |
| Seamless | 40+ | <0.001*** | <0.001*** | 0.003** | 0.28 | 0.66 | 0.84 | — |  |
|  | S40+ | 0.13 | <0.001*** | <0.001*** | 0.001** | 0.98 | 0.13 | 0.40 | — |
|  | F 40+ | 0.71 | <0.001*** | <0.001*** | <0.001*** | 0.13 | 0.002* | 0.01* | 0.55 |

Note: C 40+: Celluloid 40+, P 40: Premium 40+, F 40+: Flash 40+

## Analysis of balls elasticity tests on different material and mechanisms

One-way ANOVA revealed a significant main effect in elasticity between nine types of tested balls ($F(8, 126) = 250$, $P < 0.05$, $\eta_p^2 = 0.929$). As shown in Fig 5 and Table 4, the multiple comparison t-test showed that maximal bounce height of new material balls were all significantly higher than that of celluloid ball. Closer inspection of the Fig 5 shows seam balls reported significantly lower value of bounce height than seamless balls.

From the data in Table 4, the difference in group was also significant, either in seam or seamless. Bounce height of Yinhe S40+ was significantly higher than that of 729 ($P = 0.01 < 0.05$); Values of Japanese brands (Butterfly A40+ and Nittaku) were all significantly higher than Chinese brands (DHS D40+, DHS DJ40+ and Double Fish V40+).

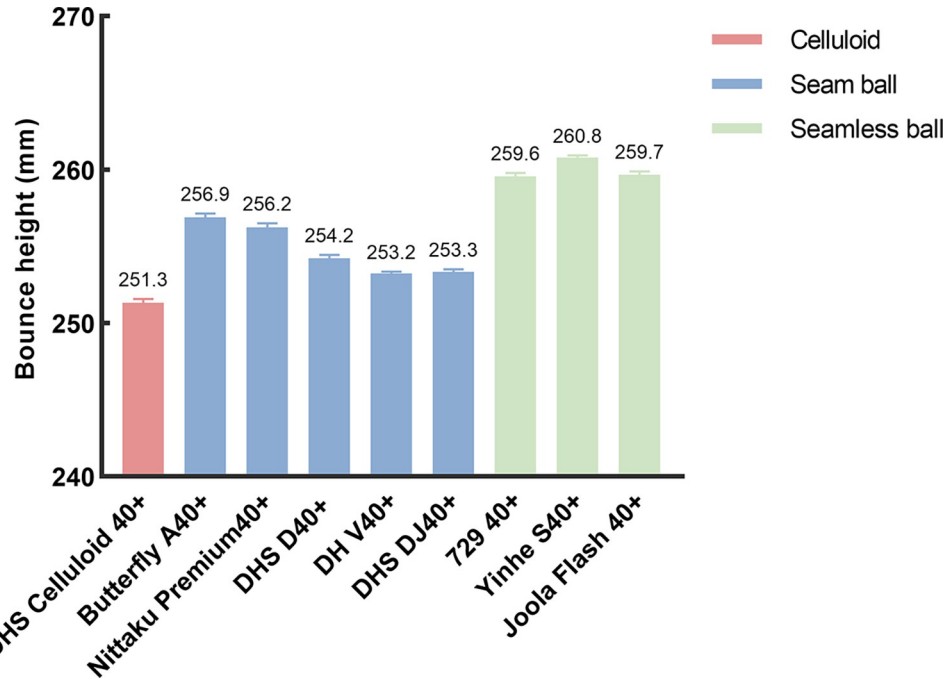

**Fig 5. Mean ± SEM of bounce height of nine types of tested balls.**

**Table 4. *P* values in Holm-Sidak's multiple comparison t-test in bounce height.**

| | Model | C 40+ | A40+ | P 40+ | D40+ | V40+ | DJ40+ | 40+ | S40+ |
|---|---|---|---|---|---|---|---|---|---|
| Seam | C 40+ | — | | | | | | | |
| | A40+ | <0.001*** | — | | | | | | |
| | P 40+ | <0.001*** | 0.05* | — | | | | | |
| | D40+ | <0.001*** | <0.001*** | <0.001*** | — | | | | |
| | V40+ | <0.001*** | <0.001*** | <0.001*** | 0.94 | — | | | |
| | DJ40+ | <0.001*** | <0.001*** | <0.001*** | 0.46 | ns | — | | |
| Seamless | 40+ | <0.001*** | <0.001*** | <0.001*** | <0.001*** | <0.001*** | <0.001*** | — | |
| | S40+ | <0.001*** | <0.001*** | <0.001*** | <0.001*** | <0.001*** | <0.001*** | 0.01* | — |
| | F 40+ | <0.001*** | <0.001*** | <0.001*** | <0.001*** | <0.001*** | <0.001*** | ns | 0.14 |

## Linear relationship between hardness and elasticity

Correlation between hardness coefficient and bounce height was analyzed to illustrate the relationship between hardness and elasticity. As shown in Fig 6A which contained the data of both seam and seamless balls, no significant correlation was found between two indicators ($R^2 = 0.646 \times 10^{-3}$, $P = 0.95$). Surprisingly, there was a significant positive correlation between hardness coefficients and bounce height ($R^2 = 0.906$, $P = 0.003$), when only considered in seam balls (Fig 6B). These results suggest that the higher hardness coefficient, the higher bounce height.

## Discussion

As mentioned in the literature review, the hardness at the equator of the seam balls is higher than that of other areas [14]. However, these studies have either been single-type studies or have not focused on differences in various details, such as structures, brands, and materials. In this study, we tested the mechanical behavior (hardness and elasticity) to which nine types of balls included seam and seamless, ABB and celluloid materials, and different brands.

We found that hardness at the equator was not always higher than that at the pole, which is contrary to previous studies. A possible explanation for this might be that an object's hardness is dependent on the material organization itself but not on the thickness [15]. Other sources of

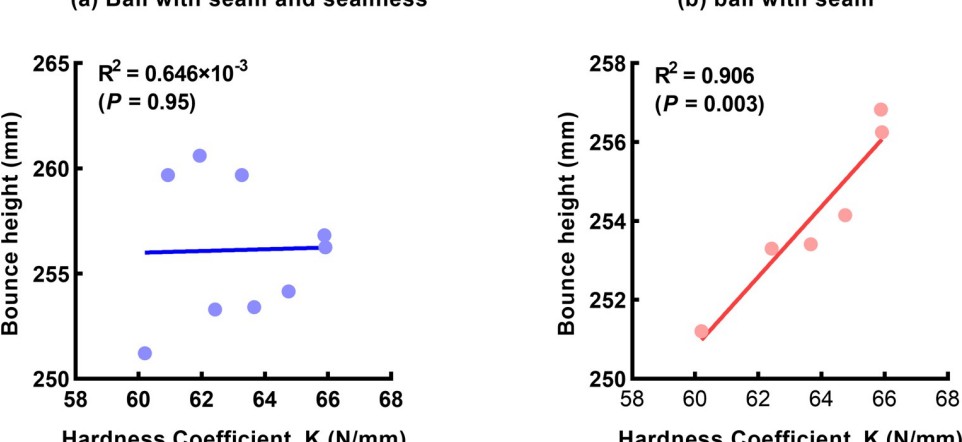

**Fig 6. Correlation between coefficient and bounce height.** (a) contained with seam and seamless balls; (b) only contained with seam balls.

uncertainty are the material formula, process differences, or the stability of the production process. It is also interesting to note that several types of balls, for example, Butterfly A40+, Nittaku Premium, and DHS D40+, have a similar hardness at the equator and pole, indicating hardness overall evenly distributed. These balls are more like the seamless balls, in which players will have better control and stable ball sense [14], without some situations like 'energetically'. Le *et al.* took into account factors such as reinforcing rib in seam balls and reported that the hardness at the striking point differs with the force angle changes [14].

As expected, this study also confirmed that the hardness and bounce height of the new material balls were greater than those of celluloid, which shows that the mechanical properties of new material balls have been improved [16, 17]. Similarly, Yuki et al. [13] calculated the post-collision trajectories of both balls by integrating the equation of motion for simulated service, smash, and drive conditions with respect to time. Based on computed variables, they found that the coefficients of restitution were higher for the plastic balls than for the celluloid ones when the initial vertical velocities were higher.

Another important result was that seamless balls have higher mechanical properties than seam balls, which further supports the finding of Xiao et al. [18]. This result may be explained by the fact that seamless balls have identical thickness and complete structure, which led to more stability of bounce height, striking point, and arc movement track [19]. A note of caution is due here, the counter-attack velocity and rotation of balls are affected by the ball's hardness. Higher hardness, the lower energy consumption, which led to the balls with faster velocity after impact. Therefore, it seems that the return speed of new material balls would be faster than celluloid. This result differs from the previous consensus that celluloid balls have faster speed and more stable rotation [20]. This phenomenon can be attributed to the radius and mass of the ball. The modern 40mm ball is 2mm larger and 0.2 grams heavier than its predecessor, the 38mm ball. Due to its increased cross-sectional area, it encounters greater air resistance [21]. In comparison to the 38mm ball, the larger ball's mass is distributed farther from its center, leading to a larger inertial moment and a decrease in rotation. The larger 40mm ball results in a reduction in both speed and rotation by approximately 5% to 10% [7–9]. Interestingly, increasing the diameter of the ball alone did not result in an apparent prolongation of the rally. This effect did not occur until the shortening of the games a year later (2001). This prolongation of the rally could have been due to the circumstance where the players were forced to heighten their level of attention, and also, it might have been the consequence of the fact that the new ball could be handled much easier by the players after a year of gathering experience [6]. Similarly, Li et al. also reported that the increased size of the ball does not fundamentally alter the characteristics of table tennis, as the greater force exerted by athletes compensates for the size increase [8].

One unanticipated finding was that the hardness of 3 kinds of seam balls (Butterfly A40+, Nittaku Premium, and DHS D40+) was much greater than the seamless balls. This outcome is contrary to previous studies. Xiao et al. [18] captured the instant deformation at ball impact using a high-speed camera. They found the seamless ball deformed earlier at ball impacting than the seam balls, indicating seamless balls were harder than the seam. This result was also consistent with Peng [22]. who pressed hard on the ball. This discrepancy could be attributed to the test modes and sample size of the selected balls. The ball's total deformation during loading includes two parts. At relatively small deformation, a linear relationship dependent on the elastic modulus of the material is found between strain and loading stress. With increasing pressure, the deformation would be nonlinear changed due to the construction and materials [23]. According to the Chinese national standard, the hardness was considered as the value of compression deformation when under loading with 50 N, which was similar to the linear regression method suggested in this study. Dynamic uniaxial compression used in the T3 [2] is

a continuous loading process, which differs from the contact time and impact force on the balls during an actual game. The instantaneous impact force might be higher than the ball's yield force, which does not result in the deformation process considered in our test.

In addition, the hardness of seamless balls in our study showed a significant correlation to the elasticity. These results indicated that the harder seamless balls have a better bounce height. However, no significant correlation was found in between the seam balls. This discrepancy may be due to the uniform structure and better roundness of seam balls.

Therefore, compared with celluloid, there are significant differences in hardness and elasticity, either the seam or seamless balls. These findings may help us to understand the effect of ball-changing on daily training and competitive game play. Zhou et al. [24] reported that increased bounce height and lower spin of the new material balls changed the manner of the serving ball, players were not always able to serve short-low backspin ball, which made the rival quickly seize the initiative. With extensive use of new material balls, surveys such as that conducted by Xu [21] have shown that Techniques and tactics in the first four rounds were complicated and varied, but became more homogeneous in subsequent rounds. Xiao et al. [18] found that new material balls have a slower speed and improved elasticity than celluloid balls. Therefore, it is possible that the time-consuming of every round increased. Players were suggested to improve energy reserves for long-time loading training and games. This finding was also reported by Zhang *et al.* [11]. The use of a new material ball has reduced the attacking success average in the first three board rounds, and increased the stalemate phase in the game, which may narrow the gap between European players and Chinese players to a certain extent.

However, far too little attention has been paid to the difference in balls performance during table tennis competitions. According to our findings, any changes in the performance of a ball likely affect the flight trajectory, thereby changing the situation of games. Recently, seam balls with new materials have been used widely in international events, but seamless balls are still allowed in ITTF tournaments. Therefore, attention to both balls is still needed. Li et al. [8] reported that after the implementation of new regulations in 2000, athletes experienced more consecutive confrontations (stalemates) than before, with the use rate of serve and return attacks remaining the same. Djokić et al [25] analyzed the differences in the game caused by rule changes in general, particularly between winning and losing players. They found that increasing the diameter of the ball reduced its speed and spin, potentially impacting the number of ace strokes and leading to fewer overplay situations, while the number of normal strokes in a rally increased. Advancements in technique underscore the increasing importance of individual players' physical fitness. In contemporary table tennis, hitting power is derived not solely from the arms but also from the coordination of the entire body. Athletes with stronger physical conditioning can employ their legs more effectively to generate greater force on the ball, compensating for the increase in size. Moreover, efficient wrist usage is necessary to generate spin. As the duration of individual sports events increases, the demand for the application of greater force on a larger scale heightens, amplifying the potential for technical errors [26].

## Conclusions

This combination of findings provides some support for daily training. Players need to improve ball control and striking stability, strengthen extraordinary explosive power, and the ability to hold the ball in the middle and far table. In addition, with the more excellent elasticity of new balls, the player must pay particular attention to regulating the height of the serving ball to avoid being passive. Coaches should also focus on official balls to make more tailored technique and tactic strategies due to the ball's performance. The study is limited by the lack of

information on process formulations and materials on each type of ball, which may help us understand the material-to-celluloid differences. An additional uncontrolled factor is a possibility that the loading method in the hardness test differs from the ball's stress condition in the actual game situation. This may lead to the subjective impression of players may be different from the results, which would be a fruitful area for further work by using the finite element model to analyze the stress condition. Simultaneously, the subjective impression of different player levels is also crucial for table tennis research.

## Supporting information

**S1 Data. Raw data shown in Fig 3.**
(XLSX)

**S2 Data. Raw data shown in Fig 4.**
(XLSX)

**S3 Data. Raw data shown in Fig 5.**
(XLSX)

**S4 Data. Raw data shown in Fig 6.**
(XLSX)

## Author Contributions

**Data curation:** Jun Wang.

**Methodology:** Jie Ren.

**Supervision:** Yan Wang.

**Writing – original draft:** Yunfei Lu.

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
