## [Decision Letter · Decision Letter 0]

16 Jan 2024

PONE-D-23-38766Effect of table tennis balls with different materials and structures on the hardness and elasticityPLOS ONE

Dear Dr. Wang,

Thank you for submitting your manuscript to PLOS ONE. After careful consideration, we feel that it has merit but does not fully meet PLOS ONE’s publication criteria as it currently stands. Therefore, we invite you to submit a revised version of the manuscript that addresses the points raised during the review process.

 Please submit your revised manuscript by Mar 01 2024 11:59PM. If you will need more time than this to complete your revisions, please reply to this message or contact the journal office at plosone@plos.org. Please include the following items when submitting your revised manuscript:A rebuttal letter that responds to each point raised by the academic editor and reviewer(s). You should upload this letter as a separate file labeled 'Response to Reviewers'.A marked-up copy of your manuscript that highlights changes made to the original version. You should upload this as a separate file labeled 'Revised Manuscript with Track Changes'.An unmarked version of your revised paper without tracked changes. You should upload this as a separate file labeled 'Manuscript'.

We look forward to receiving your revised manuscript.

Kind regards,

Emiliano Cè

Academic Editor

PLOS ONE

Journal Requirements:

Additional Editor Comments:

**ACADEMIC EDITOR:**Dear Authors,your manuscript has been reviewed by an expert in the filed that retrieved some issues you should consider wile revising the work.

Reviewers' comments:

Reviewer's Responses to Questions

**Comments to the Author**

1. Is the manuscript technically sound, and do the data support the conclusions?

Reviewer #1: Yes

2. Has the statistical analysis been performed appropriately and rigorously? 

Reviewer #1: Yes

3. Have the authors made all data underlying the findings in their manuscript fully available?

Reviewer #1: Yes

4. Is the manuscript presented in an intelligible fashion and written in standard English?

Reviewer #1: Yes

5. Review Comments to the Author

Reviewer #1: The study is very interesting with parameters that provide great information. However, some methodological issues should be added.

1. Keywords. It is suggested that the keywords be different from those in the title. Shorter words should also be included.

2. Introduction. It should be improved by briefly commenting on the implication that the characteristics of some or other balls have on the game of table tennis so that non-experts in this sport know the interest of this research.

3. Material and methods. In the samples include the characteristics of each ball analyzed in terms of mass, weight, rebound height on the table and others of interest to be able to carry out a more exhaustive contrast analysis.

4. Discussion. Delete the dot in front of the last bibliographic reference number 9 of the second paragraph.

Where "Therefore, it seems that the return speed of new material balls would be faster than celluloid" is indicated, that statement should be explained with more studies that highlight it since it generates controversy with current table tennis.

The phrase "It is difficult to explain this result, but it should be related to diameter size." It should be justified with more studies since there is extensive literature on table tennis where this fact is indicated.

What does this type imply differences in the balls on the dynamics of the game, the physical condition and the physiological response? greater or lesser demand? greater or lesser number of hits per play? Perhaps this is one of the strong points that should be highlighted in the study.

References. A review of the literature should be done since there are studies outside of Chinese publications where these issues and their effects are addressed, so it is recommended to incorporate new references to improve the introduction, discussion and bibliography, for example https://www .ncbi.nlm.nih.gov/pmc/articles/PMC5304273/

6. PLOS authors have the option to publish the peer review history of their article (what does this mean?). If published, this will include your full peer review and any attached files.

Reviewer #1: No

---

## [Author Response · Author response to Decision Letter 0]

27 Feb 2024

Dear Emiliano Cè,

Thank you for your guidance and valuable advice. In response to your requests, we have made the following revisions:

1.We have corrected our manuscript according to the template requirements of PLOS ONE.

2.Following the suggestion from PLOS ONE, we have uploaded the relevant raw data to Zenodo. The DOI for our dataset is 10.5281/zenodo.10718491. We believe that by sharing our dataset openly, we not only increase the transparency of our work but also facilitate further research and collaboration within the academic community.

3.We have uploaded the relevant raw data.

We appreciate the opportunity to improve our manuscript and hope that these adjustments meet the journal's requirements. Should you need any further information or clarification, please do not hesitate to contact us.

Sincerely,

Yan Wang

Reviewer(s)' Comments to Author:

Reviewer 1：

The study is very interesting with parameters that provide great information. However, some methodological issues should be added.

1. Keywords. It is suggested that the keywords be different from those in the title. Shorter words should also be included.

Reply: Thanks for your advice. We have changed the Keywords: Table tennis balls; Material; Hardness and Elasticity; Static mechanical characteristics. 

2. Introduction. It should be improved by briefly commenting on the implication that the characteristics of some or other balls have on the game of table tennis so that non-experts in this sport know the interest of this research.

Reply: Based on your comments, we have made revisions to enhance the clarity of our research. Specifically, we have added the impact of ball speed reduction on the spectator experience of table tennis, particularly in light of the declining media interest in the sport, particularly outside of Asia. We have emphasized the challenges viewers face in tracking the ball due to the fast-paced nature of the game, as noted by Djokić [1]. Furthermore, we have expanded on the transition from the old 38-mm ball to the larger 40-mm ball, highlighting the resultant decrease in velocity and spin, as documented by [2-4].

3. Material and methods. In the samples include the characteristics of each ball analyzed in terms of mass, weight, rebound height on the table and others of interest to be able to carry out a more exhaustive contrast analysis.

Reply: Revised. We added the diameters of each ball in Table 2.

4. Discussion. Delete the dot in front of the last bibliographic reference number 9 of the second paragraph.

Reply: We thank the reviewer for careful evaluation and pointing out these errors. We have corrected these as suggested.

5. Where "Therefore, it seems that the return speed of new material balls would be faster than celluloid" is indicated, that statement should be explained with more studies that highlight it since it generates controversy with current table tennis. The phrase "It is difficult to explain this result, but it should be related to diameter size." It should be justified with more studies since there is extensive literature on table tennis where this fact is indicated. 

Reply: Thank you for your insightful comments regarding our paper. We greatly appreciate your feedback and have carefully considered your suggestions. As you rightly pointed out, the changes observed in the speed of table tennis balls generates controversy with current table tennis. 

The phenomenon of decreased speed and rotation in table tennis balls can be attributed to the changes in the radius and mass of the ball. The modern 40mm ball, being 2mm larger and 0.2 grams heavier than its predecessor, encounters greater air resistance due to its increased cross-sectional area. This, in turn, affects its speed and rotation. Additionally, the redistribution of mass away from the center of the larger ball results in a larger inertial moment, further diminishing rotation. Furthermore, research by Bai et al. [4], Iimoto et al. [2], and J. Li et al. [3] supports our findings, indicating that the larger 40mm ball leads to a reduction in both speed and rotation by approximately 5% to 10%.

Furthermore, we added the impact of ball radius variation on player performance. The increase in ball diameter not only affects the characteristics of the game but also influences players' strategies and techniques. Players need to adapt their playing style to accommodate the changes in ball dynamics, which can significantly impact their performance on the court [5, 6].

6. What does this type imply differences in the balls on the dynamics of the game, the physical condition and the physiological response? greater or lesser demand? greater or lesser number of hits per play? Perhaps this is one of the strong points that should be highlighted in the study.

Reply: We appreciate your feedback and have carefully reviewed your suggestions. Regarding your question about material changes' impact on player performance, we'd like to offer further explanation based on additional literature we've included. As per Li et al. [3], new regulations in 2000 significantly altered table tennis dynamics. Athletes faced more stalemates, while serve and return attack rates remained constant. This highlights the direct influence of ball materials and regulations on player strategies.

Additionally, Djokić et al. [6] support our findings, showing that increasing ball diameter reduces speed and spin, affecting ace strokes and overplay situations. Their research emphasizes the importance of player fitness and technique in adapting to these changes.

7. References. A review of the literature should be done since there are studies outside of Chinese publications where these issues and their effects are addressed, so it is recommended to incorporate new references to improve the introduction, discussion and bibliography, for example: 

 https://www .ncbi.nlm.nih.gov/pmc/articles/PMC5304273/

Reply: We have carefully reviewed the provided link to the study on the effects of the issues discussed in our paper. We also added some references to enrich the discussion.

References:

1. Djokić, Z. ITTF scored a goal (changes of rules in table tennis during 2000-2003). in Proceedings book. 10th International Table Tennis Sports Science Congress. 2007.

2. Iimoto, Y., K. Yoshida, and N. Yuza, Rebound characteristics of the new table tennis Ball; Differences between the 40 mm (2.7 g) and 38 mm (2.5 g) balls. Int J Table Tennis Sci, 2002. 5: p. 233-243.

3. Li, J., X. Zhao, and C. Zhang, Changes and development: Influence of new rules on table tennis techniques. Sports Science Research, 2005. 26(3): p. 55.

4. Bai, K., et al. Technical contrastive analysis after ping-pong diameter altering. in Proceedings of the 9th ITTF Sports Science Congress, Shanghai, China. 2005.

5. Pradas, F., et al., Analysis of Specific Physical Fitness in High-Level Table Tennis Players-Sex Differences. Int J Environ Res Public Health, 2022. 19(9).

6. Đokić, Z., et al., Effects OF rule changes ON performance efficacy: differences between winners and losers table tennis players. Facta Universitatis, Series: Physical Education and Sport, 2019: p. 149-163.

---

## [Editor Report · Decision Letter 1]

19 Mar 2024

Effect of table tennis balls with different materials and structures on the hardness and elasticity

PONE-D-23-38766R1

Dear Dr. Wang,

We’re pleased to inform you that your manuscript has been judged scientifically suitable for publication and will be formally accepted for publication once it meets all outstanding technical requirements.

Kind regards,

Emiliano Cè

Academic Editor

PLOS ONE

Additional Editor Comments (optional):

The Authors replied adequately to all the points raised by the reviewer. No further revision is required
---

## [Editor Report · Acceptance letter]

24 Mar 2024

PONE-D-23-38766R1 

PLOS ONE

Dear Dr. Wang, 

I'm pleased to inform you that your manuscript has been deemed suitable for publication in PLOS ONE. Congratulations! Your manuscript is now being handed over to our production team.

Kind regards, 

on behalf of

Prof. Emiliano Cè 

Academic Editor

PLOS ONE